# A New Noise Shaping Approach for Sigma-Delta Modulators Using Two-Stage Feed-Forward Delays and Hybrid MASH-EFM

Khalid Ijaz [1], Muhammad Adnan [2], Waqas Tariq Toor [3], Muhammad Asim Butt [1], Muhammad Idrees [4], Usman Ali [5], Izaz Hassan [6], Yazeed Yasin Ghadi [7], Fuad A. Awwad [8], Mohamed R. Abonazel [9,*] and Syed Rehan Ashraf [10]

1. Department of Electrical Engineering, University of Management and Technology, Lahore 54770, Pakistan
2. Department of Computer Science, University of Management and Technology, Lahore 54770, Pakistan
3. Department of Electrical Engineering, University of Engineering and Technology Lahore, Lahore 39161, Pakistan
4. Department of Computer Science, University of Engineering and Technology Lahore, Lahore 39161, Pakistan
5. Department of Electrical and Computer Engineering, Sungkyunkwan University, Suwon 03063, Republic of Korea
6. Department of Software Engineering, University of Science and Technology, Bannu 28100, Pakistan
7. Department of Computer Science, Al-Ain University, Al Ain P.O. Box 64141, United Arab Emirates
8. Department of Quantitative Analysis, College of Business Administration, King Saud University, P.O. Box 71115, Riyadh 11587, Saudi Arabia
9. Department of Applied Statistics and Econometrics, Faculty of Graduate Studies for Statistical Research, Cairo University, Giza 12613, Egypt
10. Department of Industrial Engineering, University of Management and Technology, Lahore 54770, Pakistan
* Correspondence: mabonazel@cu.edu.eg

**Abstract:** Sigma-delta modulators use a noise-shaping technique to curtail the noise power in the band of interest during digital-to-analog conversion. Error feedback modulator employs an efficient noise transfer function for time varying inputs than any other sigma-delta modulators. However, the efficiency of the conventional noise transfer function degrades and the quantizer saturation issue provokes when the input signal reaches to full scale. This work proposes a new noise transfer function which is a combination of transfer functions of two-stage Feed-forward delays and a novel Hybrid multi-stage noise shaping-error feedback sigma-delta modulator. The noise transfer function of two-stage Feed-forward delays mitigates the concern of quantizer saturation. The noise transfer function offered by the Hybrid multi-stage noise shaping-error feedback architecture provides sustainable solutions to limit cycles and idle tones. The simulation concludes that the proposed noise-shaping approach obtains comparatively high signal-to-quantization noise ratio than the conventional error feedback modulators. Other performance parameters like spurious-free dynamic range, effective number of bits and signal-to-noise plus distortion ratio are also significantly improved.

**Keywords:** sigma-delta modulation; digital-to-analog converter; noise shaping; quantization error feedback modulator

## 1. Introduction

High resolution analog-to-digital converters (ADCs) and digital-to-analog converters (DACs) extensively use sigma-delta modulation. This technique conveniently provides high signal-to-quantization noise ratio (SQNR), which makes it important to use in super audio CD format. It is widely used in recent data acquisition systems, Microelectromechanical systems (MEMS), inertial sensors, image sensors, wireless technologies such as long-term evolution advanced (LTE-Advanced), global system for mobile Telecommunication (GSM) and code-division multiple-access (CDMA) etc., where high speed ADCs with wide band-width, low noise and less glitch-induced harmonic distortion are required [1–9]. Portable audio playback devices, which perform digital-to-analog conversion also make use of it.

Other applications include instrumentation, seismic activity measurements, speech, video, Integrated System digital Network (ISDN), digital cellular radio, mobile communication devices and systems by reducing the complexity of sigma-delta modulators through different designs considerations, next generation wireless telecom systems, frequency synthesizer, Chromatography and electrocortical reading in brain stimulation applications [10–14]. A quantization noise is generated during analog-to-digital and digital-to-analog conversion, which is the main cause to accomplish reconstruction process inappropriate [15]. Sigma-delta modulator (SDM) employs noise shaping and oversampling, which move this noise from the band of interest to higher frequency region [16].

Sigma-delta modulator comprises of "sigma" and "delta" blocks [17]. The input is passed through a feed forward filter $\left(\frac{z^{-1}}{1-z^{-1}}\right)$, which is a sigma block followed by a quantizer. If two feed forward filters are used in two stages, then it is called second order sigma-delta modulator [18]. Delta block subtracts output from the input [19,20]. The feed forward filter combined with delta block is termed as a loop filter [21]. The output can be generalized in Equation (1):

$$Y(z) = STF(z)X(z) + NTF(z)E(z) \tag{1}$$

In previous research work, Multi-stage noise shaping (MASH) based SDM was widely used for high-bandwidth applications. In MASH architecture, the input is applied to the integrator (feed forward filter), which comprises of accumulator and delay blocks. The signal reaches to the quantizer after passing through the feed forward filter. The quantized output is then fed back to the delta block to subtract it from the input. The average output of the quantizer follows the average input through feedback path. However, the MASH architecture does not provide stability with time varying inputs and eventually degrades its performance. In Error feedback modulator (EFM), the quantizer saturates and overflows when the input signal is closed to full scale. The other main issue observed in 1-bit EFM is the existence of limit cycles in the frequency spectrum. In limit cycles, the output bits repeat indefinitely, and the output spectrum of SDM goes into periodic modes. If the output bit stream of SDM remains periodic and occurs after a short period, then it gives rise to short limit cycles. Due to which, strong unwanted peaks occurred in the spectrum. While in conventional hybrid models, the mapping of discrete/digital time transfer functions to the continuous time transfer functions is complex. Hence, the major issue observed in the designing of SDM is stability with low performance parameters like low SQNR and occurrence of limit cycles.

This paper proposes a new architecture named as Hybrid Multi-stage noise shaping error feedback modulator (MASH-EFM), which uses some major characteristics of both MASH and EFM architectures. Feed forward filters and error signal, which are the main attributes of MASH and EFM architectures respectively, are used in this architecture. Limiters and dithering avoid overflow and remove limit cycles, respectively. Consequently, this architecture is more stable and robust. It provides high SQNR for both time varying and non-time varying inputs and mitigates all the issues discussed in the above paragraph. Moreover, this proposed hybrid architecture seeks to eliminate the mapping of discrete time transfer function to the continuous time transfer function.

This work emphasizes on the development of a third order Hybrid MASH-EFM that can be suitable for 1-bit audio DAC applications due to its better noise shaping. As the occurrence of limit cycles are more prominent in 1-bit EFM, therefore 1-bit architecture is proposed to show that this presented architecture effectively removes limit cycles. And hence it is superior to EFM. The other main factor to emphasize 1-bit architecture is to implement 1-bit quantizer. The complexity of quantizer increases as the number of bits increases, which results in the increase of cost. This proposed architecture is designed in such a way that it provides cost effective solution with high SQNR. Two-stage Feed forward digital loop filters are used, which also reduce the complexity of mapping between discrete-time (DT) and continuous-time (CT) blocks in a Hybrid architecture. Moreover,

the application of third-order Hybrid MASH-EFM can also be extended for Video DAC by improving its resolution.

This article is organized as follows. Section 2 mentions the research work that has already been carried out to develop different variants of MASH and EFM sigma-delta modulators. Section 3 describes the block diagram of proposed architecture and mathematically derives the important $\sum\Delta$ parameters like noise transfer function (NTF), SQNR and effective number of bits (ENOB). In Section 4, simulation methodology has been discussed for conventional and proposed architectures. All the simulations have been performed in this section. Results have been explained in both graphical and tabular form and comparison between proposed and existing architecture have been deduced in Section 5. Section 6 represents the conclusions and future recommendations of this paper.

## 2. Related Work

A lot of work has been published on MASH, EFM and different types of hybrid SDM architectures. MASH can be easily constructed and implemented with greater accuracy delivering high performance parameters because of its aggressive noise shaping [22,23]. However, MASH architecture loses stability and performance during time-varying inputs. Cascaded MASH architectures are also implemented several times in different research papers. The cascaded MASH architectures were complex and complicated in terms of circuitry.

The above stability and complex circuitry issues are removed by the EFM architecture. In EFM, the error signal is obtained by subtracting the output of the EFM with the input of the quantizer and then given back to the input signal. EFM uses error signal and loop filter in the feedback path. The NTF of Nth order EFM is $\left(1 - z^{-1}\right)^{N}$ [24]. So, the third order NTF is given by: $\left(1 - z^{-1}\right)^{3}$. The EFM architecture provides greater stability with time varying inputs [25–27]. The EFM architecture is shown in Figure 1. In Figure 1, $x_d$ represents the input to the EFM, $e_i$ is the error signal generated by subtracting the input signal $x_d$ and feedback signal $e_h$, and Q(.) denotes the quantizer. The output of the quantizer $y_d$ is converted into digital signal y by the DAC. However, the EFM architecture saturates when the input reaches to full scale.

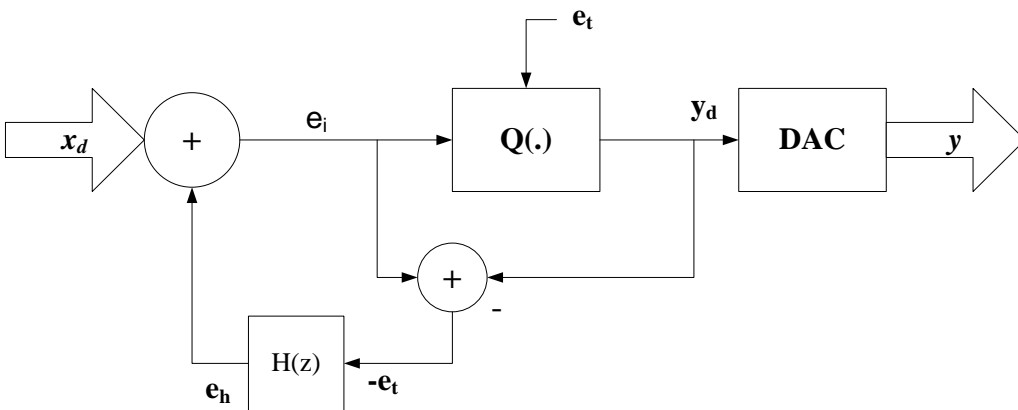

**Figure 1.** Error Feedback Modulator (EFM) Architecture.

The Hybrid audio SDM was also presented to resolve the saturation issue of the quantizer [15,28]. The Hybrid audio SDM consists of analog integrator, successive approximate registers (SAR), ADC, digital filter with excessive loop delay (ELD) compensator, a feedback DAC, and a dynamic element matching (DEM) block. Analog integrator is placed in the first stage and the second stage composes of the digital filters, DEM, and DAC. The output of an analog integrator is converted into digital form by SAR. A bit by bit conversion is performed by the left channel when right channel performs sampling and vice versa. Other type of hybrid SDM in which continuous and discrete time transfer functions are cascaded was also implemented [29]. The hybrid SDM based on multi-rate down sampling,

named as "multi-rate down sampling Hybrid continuous-time (CT)/discrete-time (DT) cascaded SDM", was also presented by the researchers [30]. Moreover, the mapping of CT and DT transfer function is complicated as the DT signal, which is the output of the quantizer is fed back to CT block. The above-mentioned drawback can be removed by using the digital filters as the input of the feedforward filter as first time introduced in the proposed architecture.

### 3. A New Proposed Noise Transfer Function

The major characteristics of aforementioned existing architectures are incorporated to create a new architecture, which shows better efficiency than the existing architectures.

The block diagram of third-order Hybrid MASH-EFM is shown in Figure 2. The two sets of useful delays or digital filters $Z^{-2}$ and $Z^{-1}$ are used in this architecture indicated as "delay set 1 and delay set 2" respectively. The feed forward filters (feed forward filter 1 and feed forward filter 2), which are the characteristics of MASH, are placed in the feed forward path in both the stages respectively. Similarly, error signal, which is the characteristic of EFM, is extracted by subtracting the output of a proposed SDM with the input of a quantizer. Limiter 1 and limiter 2 are also deployed in this feedback path. Limiter 1 improves noise shaping and avoids overflow when the input signal is closed to full scale and quantization error becomes unbounded. Limiter 2 also prevents it from an overflow by completely suppressing the feedback signal. Limiter 1 and limiter 2 are also indicated in Figure 2. $K_a$, $K_b$, $K_c$, $K_d$ and $K_e$ are the gains. The output spectrum of proposed architecture is a spur-free at a higher gain, which shows that proposed architecture is more robust and stable due to limiter circuits.

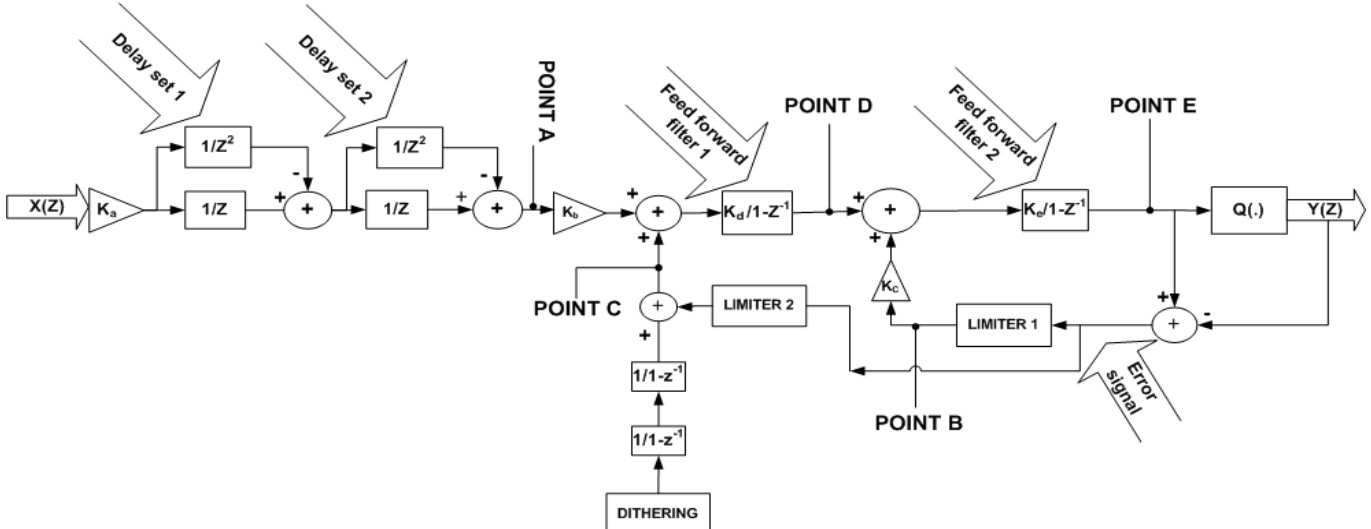

**Figure 2.** A New Proposed MASH-EFM architecture.

A low power random dither sequence denoted by d[n] is applied to this architecture. The output of a dither block, which is shown in Figure 3, can be mathematically written in Equation (2):

$$o[n] = d[n] * V_1(z)V_2(z) + l3[n] + u[n] \qquad (2)$$

where "$*$" is convolution operator. "$V_1$ (z)" and "$V_2$ (z)" are the low pass filters, and both have transfer functions given by $\frac{1}{(1-Z^{-1})}$. $l3[n]$ and $u[n]$ are the output signals in the discrete-time domain from the limiter and two-stage feed-forward delays respectively. In higher order modulator, dithering makes the noise distribution uniform, reduces the tones, and removes limit cycles by breaking the sequence of repetitive bit stream at the output of quantizer. Usually, the minimum cycle length of SDM is very small. The overall power spectrum of quantization noise also consists of power of dither signal.

POINT A

**Figure 3.** Dithering in proposed Architecture.

For simplicity, the quantization error $E_1(z)$ can be generically replaced by the notation $E(z)$, as written in Equation (3):

$$ErrorSignal = E_1(z) = E(z) \tag{3}$$

The output after two set of delays or at point "$A$" is mathematically expressed in Equation (4):

$$Signal\ at\ A = K_a K_b X(z) Z^{-2} \left(1 - Z^{-1}\right)^2 \tag{4}$$

The output of limiter 1 is mathematically represented in Equation (5):

$$Signal\ at\ B = E(z)\left(1 - Z^{-1}\right)^4 \tag{5}$$

If the Quantizer is replaced by its equivalent model, then the output $Y(Z)$ will be written in Equation (6) as:

$$Y(z) = K_a K_b K_d K_e X(z) Z^{-2} + D(z) + K_c K_d K_e E(z)\left(1 - Z^{-1}\right)^3 \tag{6}$$

where $D(z)$ represents the dither signal in z-domain.

By setting, $K_a K_b K_d K_e = K_s$, and $K_c K_d K_e = K_n$. Hence, the signal transfer function (STF) and NTF can be mathematically delineated in the form of Equations (7) and (8) respectively.

$$STF(z) = K_s Z^{-2} \tag{7}$$

$$NTF(z) = K_n \left(1 - Z^{-1}\right)^3 \tag{8}$$

The gains are adjusted in such a way that $K_n$ should not be greater than 4 to reduce the in-band noise gain. The third-order Hybrid MASH-EFM architecture acquires same NTF as 3rd order EFM SDM. Hence, this architecture acts as a 3rd order noise shaper.

The 3rd order NTF of proposed architecture is mathematically expressed in Equation (9):

$$NTF(z) = 4\left(1 - Z^{-1}\right)^3 \tag{9}$$

The mathematical expression for squared magnitude of NTF is derived by replacing $Z$ to $e^{j\omega}$ as given in Equation (10):

$$\left|NTF\left(e^{j\omega}\right)\right|^2 = |4|^2 \cdot \left|\left(1 - e^{-j\omega}\right)^3\right|^2 \tag{10}$$

The SQNR can also be determined through the NTF. It is calculated by dividing the original signal power with an in-band, shaped quantization noise power at the frequency of an input signal. Assume that the power of input signal and the noise power is represented by $\sigma_x^2$ and $\sigma_n^2$ respectively. The quantization noise power is denoted by $\sigma_e^2$. Then, the power of the input signal and the noise power can be mathematically expressed in Equations (11) and (12) as:

$$\sigma_x^2 = \frac{A^2}{2} \tag{11}$$

where $A$ represents the amplitude of the input signal.

$$\sigma_n^2 = \int_{-f_B}^{f_B} |NTF(\omega)|^2 \sigma_e^2 d\omega \tag{12}$$

The sampling frequency is mapped to $2\pi$. The oversampling ratio (*OSR*) also have a greater impact on the noise power, thus the noise power can be expressed in terms of Equations (13) and (14) as:

$$\sigma_n^2 = 2 \cdot \frac{1}{2\pi} \sigma_e^2 \cdot 16 \int_0^{\frac{\pi}{OSR}} 2^6 \left[ \sin^3\left(\frac{\omega}{2}\right) \right]^2 d\omega \tag{13}$$

$$\sigma_n^2 = 16\sigma_e^2 \cdot \frac{\pi^6}{7OSR^7} \tag{14}$$

Though, the *SQNR* is mathematically written in the form of Equation (15) as:

$$SQNR = \frac{\sigma_x^2}{\sigma_n^2} \tag{15}$$

The *ENOB*, which depends upon the *SQNR* can be defined by the Equation (16):

$$ENOB = \frac{(SQNR)_{dB} - 1.76}{6.02} \tag{16}$$

By solving the Equation (16), the *ENOB* can be expressed by the Equation (17):

$$ENOB = \frac{\sigma_x^2}{0.38 \frac{\Delta^2}{12} \frac{\pi^6}{OSR^7}} - 74.1 \tag{17}$$

where $\Delta$ and *OSR* are the step size and *OSR* respectively. According to Equation (17), the *ENOB* depends upon the step size, *OSR*, and input signal power. By increasing the quantizer bits, the step size decreases. Increasing the quantizer bits and the *OSR* are all the methods to improve the *ENOB* in the proposed architecture. But there is a trade-off between *ENOB*, *OSR* and quantizer bits because increasing the *OSR* and quantizer bits saturate the quantizer and become impractical for DAC. Therefore, this architecture uses single bit quantizer and *OSR* is set to 64.

## 4. Simulation Methodology

A Sine wave of 1 V and 20 kHz has been generated in SIMULINK. The input signal is amplified to 5 V to validate the superiority of the proposed architecture over a larger time- varying input signal. The noise signal is also added in the Then the amplified signal has been sampled at over sampling ratio of 64 [31,32]. The Power Spectral Density (PSD) and performance parameters of both EFM and the proposed two-stage Feed forward filters based Hybrid MASH-EFM architecture are compared.

### 4.1. Simulation of Second-Order EFM

The PSD of conventional EFM is displayed in Figure 4, which shows that the quantization noise is suppressed in frequency band of interest (baseband) and transferred it to higher frequency region. However, 1-bit conventional EFM is more prone to Limit cycles, indicated in dotted ellipse portion, due to the occurrence of same bit pattern repeatedly at the output of its quantizer. The simple method to detect the limit cycle is to compare the output bit stream at each successive iterate. For some iterates "*i*", the output bit stream, S(0) = S(*i*), then a limit cycle of some period exists. The Limit cycle is the major issue in the enhancement of performance of SDM [33–36]. For this purpose, the input and output bit stream as a function of iterate are plotted in Figures 5 and 6 respectively.

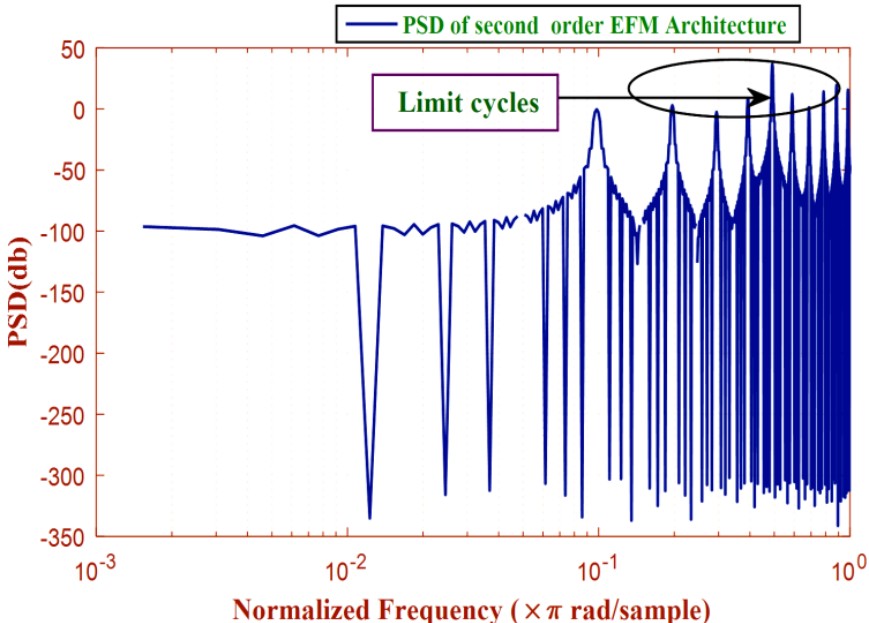

**Figure 4.** PSD of Second Order EFM with *OSR* = 64.

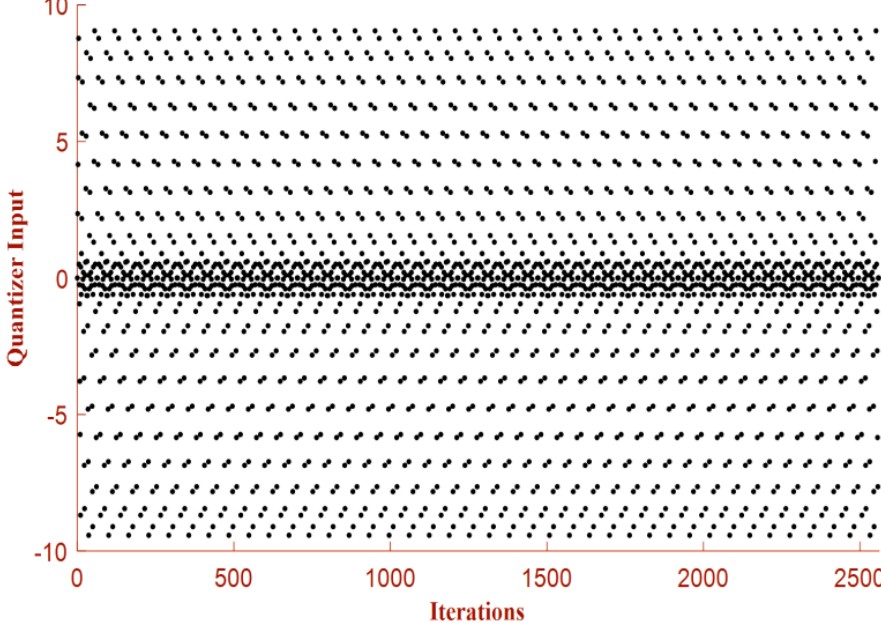

**Figure 5.** Plot of Quantizer Input as a function of iterations.

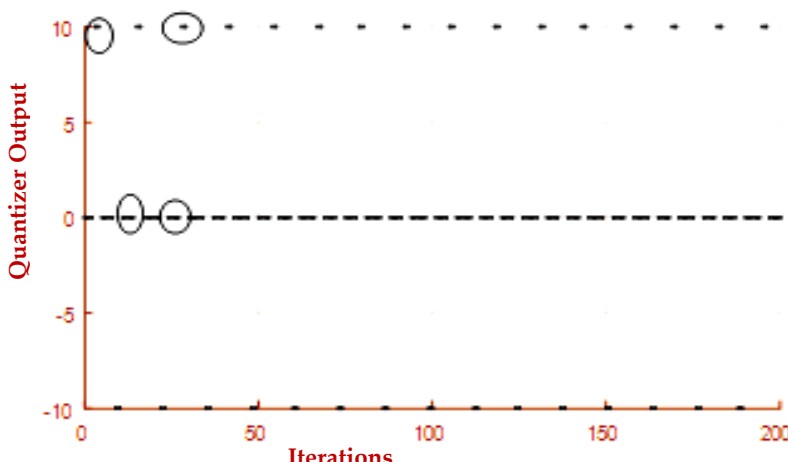

**Figure 6.** Plot of Quantizer Output as a function of iterations.

### 4.2. Simulation of Proposed Hybrid MASH-EFM Architecture

The SIMULINK model of proposed architecture is shown in Figure 7. Limiter 1 and limiter 2 prevent a proposed architecture from overflow, when high gain is applied to the proposed architecture. Figure 8 represents the PSD of the proposed architecture. The quantization noise is completely suppressed in the lower frequency region or in the DC frequency region as shown in the Figure 8. The NTF curve is plotted in black color. The NTF of proposed architecture delineates that the higher attenuation occurs at low frequency region (band of interest) and therefore the quantization noise is suppressed in this region. The Signal bandwidth is 40 KHZ and the bell-shaped curve, magnified in Figure 7, contains the useful data. Next to this frequency region, noise is transmitted due to Noise shaping. The PSD shows that quantization noise follows the NTF curve (function) of proposed architecture, which leads to the smooth spectrum. At higher frequency, limit cycles are removed due to dithering. Figure 9 also represents that the quantized output of proposed architecture does not experience the repetition of the same bits.

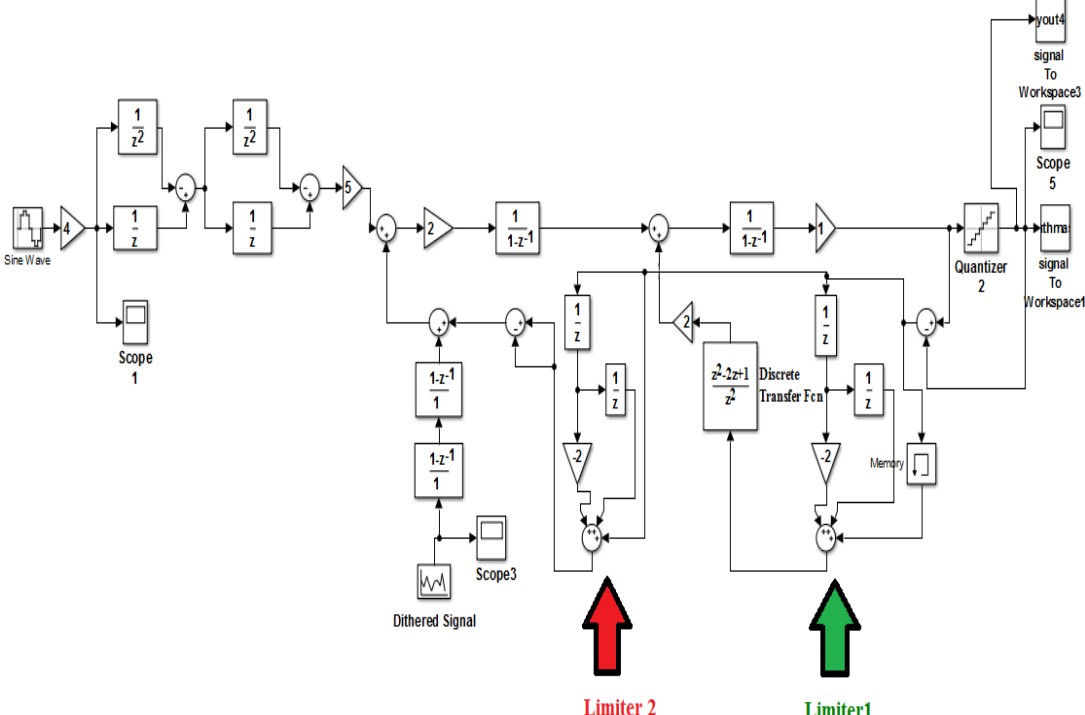

**Figure 7.** Simulink Model of Hybrid MASH-EFM Architecture.

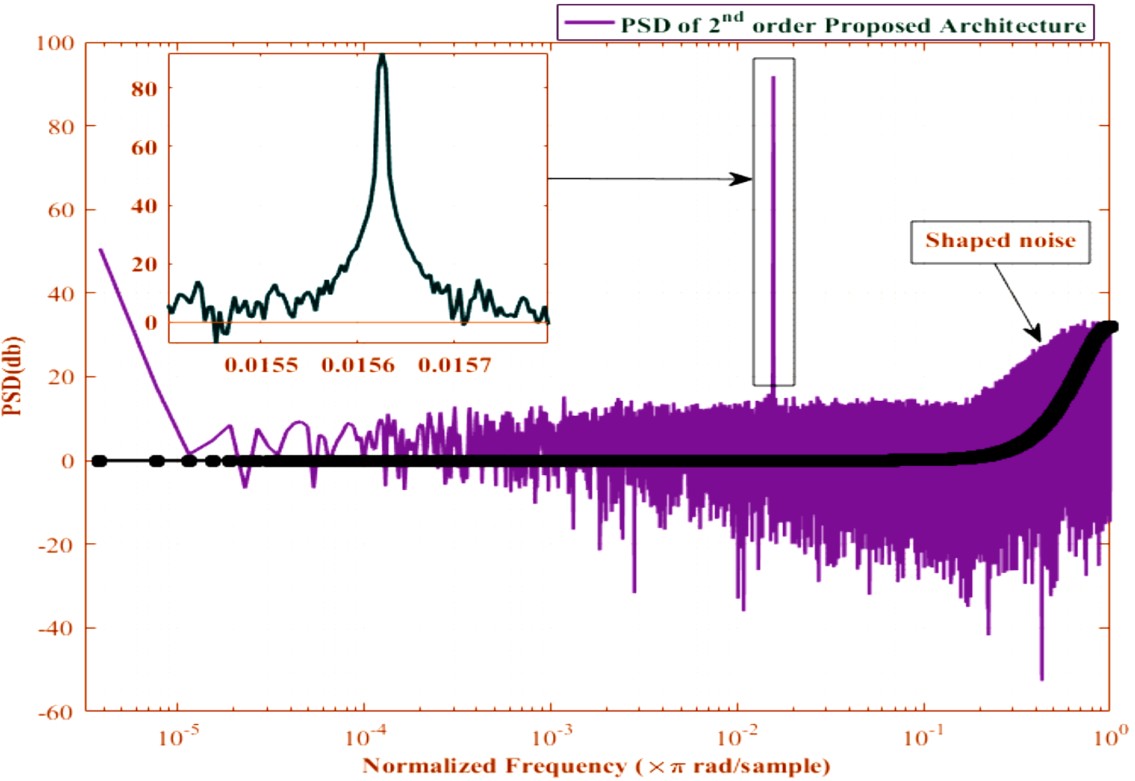

**Figure 8.** PSD of Hybrid MASH-EFM Architecture with *OSR* = 64.

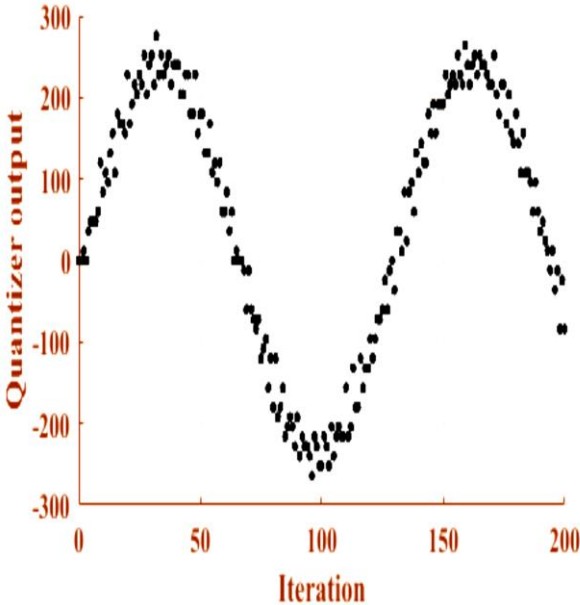

**Figure 9.** Quantizer output of Hybrid MASH-EFM.

## 5. Experimental Results and Discussion

The foremost advantage of proposed architecture is that the Hybrid MASH-EFM uses two limiters to prevent a quantizer from overflow at large input signal. The effect of quantization noise becomes worst at the larger input signal. To prevent the above-mentioned issue, the output of limiter 1 which consists of quantization noise in the feedback path is fed back to the accumulator of second stage. The limiter 2 is designed in such a way that it totally suppresses the quantization noise as described in the mathematical modeling of proposed architecture so that the error signal is not summed up with the input signal

again at the accumulator of first stage. Hence, a proposed Hybrid MASH-EFM prevents the quantizer to go into the non-linear behavior. Also, the hardware complexity and power consumption are increased due to the use of limiter circuits in third-order Hybrid MASH-EFM. But the additional increase in hardware complexity and power consumption of Hybrid MASH-EFM architecture is minor because these limiters use digital filters and delays. Hence, a proposed architecture provides greater stability and robustness at a very small cost of hardware complexity [37]. In contrast to a proposed architecture, a conventional EFM does not possess such phenomena to limit the consequences of the error signal. In conventional EFM, the error signal is then summed up with the input signal which makes the quantizer overwhelmed at a full scale of the input signal which results in its instability.

The 3rd order EFM has NTF: $\left(1 - z^{-1}\right)^3$ as discussed in Section 1. While the proposed third-order architecture achieves same NTF with its scaling factor of 4. Again, the use of limiters in the feedback path helps to obtain higher order noise shaping. Figure 10 represents the NTF graph of 2nd order EFM, 3rd order EFM and 3rd order Hybrid MASH-EFM. The black curve represents the squared magnitude of NTF of proposed architecture. While the green and blue curves represent the squared magnitude of NTF of 2nd and 3rd order EFM respectively. The NTF graph depicts that the proposed architecture will provide higher attenuation to the quantization noise in the signal bandwidth and lower attenuation at higher frequencies than the conventional 2nd order EFM architectures due to the scaling factor of NTF of Hybrid MASH-EFM. Therefore, the Hybrid MASH-EFM assists aggressive noise shaping than 2nd and 3rd order EFM. The PSD of both conventional EFM and Hybrid MASH-EFM are shown in Figures 11 and 12 respectively. The PSDs of both conventional EFM and Hybrid MASH-EFM suggest that the noise shaping is more prominent in the proposed architecture.

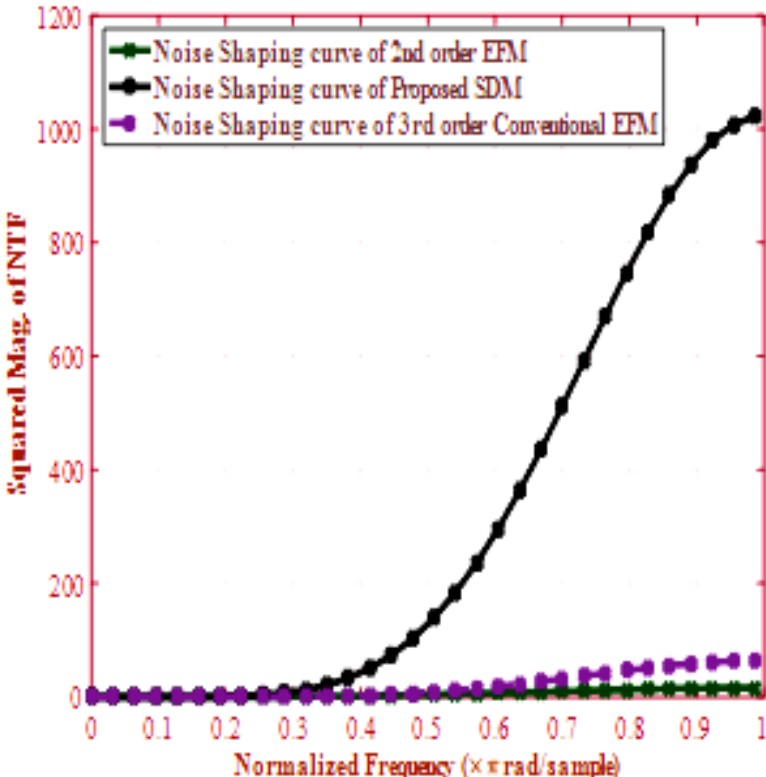

**Figure 10.** NTFs of 2nd order EFM, 3rd order EFM, and 3rd order hybrid MASH-EFM.

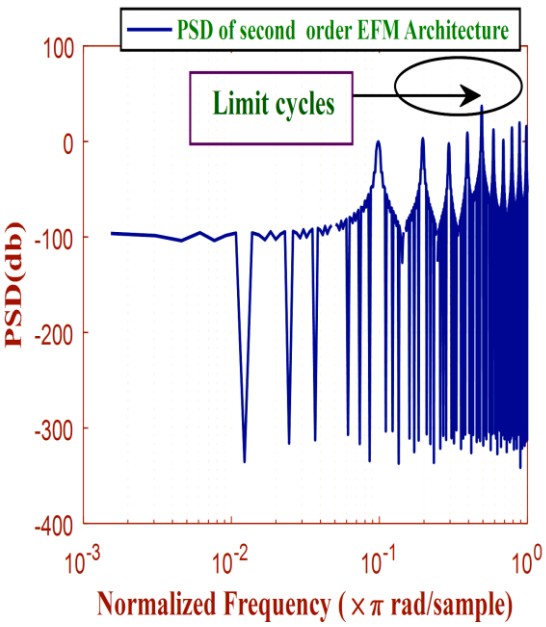

**Figure 11.** PSD of 2nd order conventional EFM.

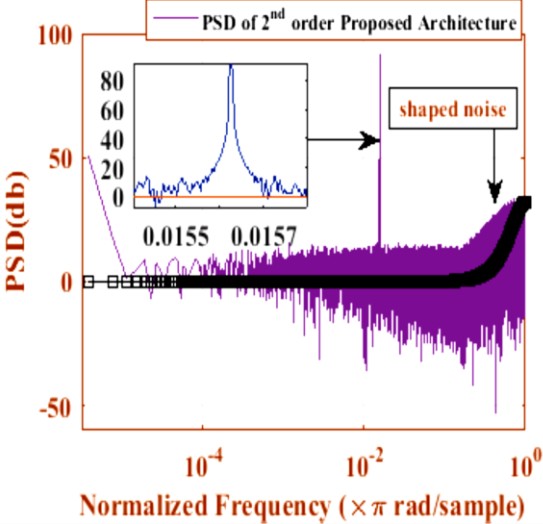

**Figure 12.** PSD of 3rd order hybrid MASH-EFM.

Moreover, some tones are also being observed at lower frequencies in EFM as presented in the PSD of 2nd order EFM in the Figure 11. In contrast, the NTF of proposed architecture makes the distribution of quantization noise smooth and free from idle tones. The dark black curve is the NTF in the PSD of proposed architecture. The quantization noise is perfectly matched with the NTF curve. The STF of proposed architecture behaves like a low pass filter as in the case of EFM. The PSD of Hybrid MASH-EFM reveals that the quantization noise is completely suppressed in the DC frequency region. The STF provides lower attenuation to the signal in the baseband and higher attenuation in the higher frequency region. As the frequency increases, the quantization noise also increases in the spectrum.

The other main cause of instability is the occurrence of limit cycles in conventional EFM. To demonstrate the above discussed issue in EFM, the graph between number of iterations and the input to the quantizer is plotted, which is already shown in the Figure 5. The scattered plot of quantizer input bit stream tells that the quantizer does not experience the recurrence of same bit. In contrast to the Figure 5, Figure 6 suggests that the duplication

of same bits is observed in the output bit stream indicated by black circles which produces limit cycles. The first 200 iterations are plotted in the Figure 6 to analyze the results clearly. The PSD of EFM proves that limit cycles are observed due to these duplications as shown in Figure 11. Thus, a 1-bit conventional EFM is more prone to limit cycles. This problem is mitigated in a proposed architecture. This work demonstrates that if the pseudorandom dither (a disturbance) is added at the output of Limiter 2 in the feedback path in 3rd order Hybrid MASH-EFM, which effectively disturbs the repetitive bit pattern at the output of quantizer. The scattered pattern of output bit stream of quantizer in Hybrid MASH-EFM determines the absence of reappearance of same bit as indicated in the Figure 9. The output bit stream of Hybrid MASH-EFM follows the pattern of input signal after removing the repetitive bits. Figure 13 represents the PSD of quantization noise with dithering in proposed architecture and Figure 14 makes comparison of quantization noise between existing and the proposed architecture. The quantization noise, which is in black curve in the Figure 14, is free from limit cycle while conventional EFM architecture experiences limit cycles in its quantization noise (in red color). Equation (9) also reveals that the application of dither signal at the output of Limiter 2 does not affect the STF and NTF, which is another main advantage. Hence the proposed architecture is more stable in order to resist limit cycles.

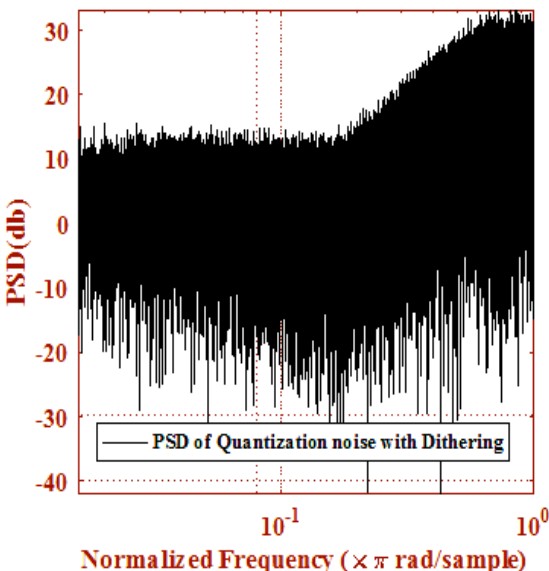

**Figure 13.** PSD of quantization noise with dithering.

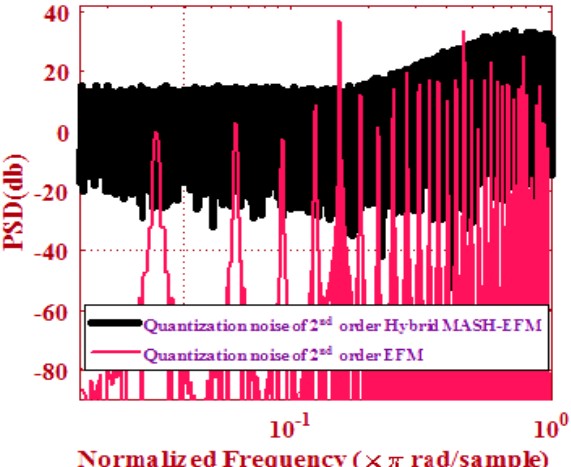

**Figure 14.** Comparison of the quantization noise of EFM and proposed architecture.

Simulation results also predict that the proposed architecture obtains higher SQNR as compared to 2nd order EFM due to aggressive noise shaping and stability as acknowledged in Table 1. The SQNR versus input signal level are also plotted for 2nd order EFM and 3rd order Hybrid MASH-EFM as displayed in Figure 15. The SQNR value in Hybrid MASH-EFM architecture reaches to 86.68 dB which is greater than EFM for the same input signal.

**Table 1.** Comparison of conventional 2nd order EFM and Hybrid MASH-EFM.

| SDM Architectures | 2nd Order EFM | 3rd Order Hybrid MASH-EFM |
|---|---|---|
| Input Signal Frequency | 20 kHz | 20 kHz |
| Signal Bandwidth | 40 kHz | 40 kHz |
| OSR | 64 | 64 |
| SQNR | 75.25 | 86.68 |
| SFDR | −36.0 | 76.53 |
| SNDR | 84.56 | 85.24 |
| ENOB | 12.20 | 14.16 |

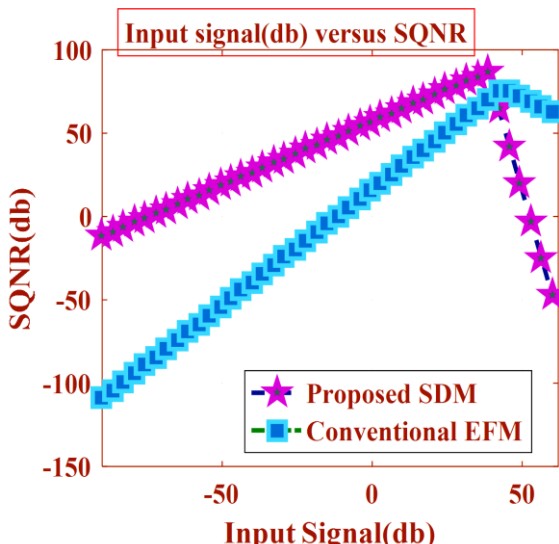

**Figure 15.** SQNR vs. Input signal.

As discussed earlier, the proposed architecture effectively mitigates the problem of limit cycle which results in the improvement of spurious-free dynamic range (SFDR). SFDR is defined as the ratio of the RMS amplitude of the digital signal to the RMS value of the strongest spurious components. In Figure 16, the red and blue curves represent the SFDR of proposed and conventional architecture respectively. The peak SFDR achieved during simulation is about 76.53 dB, which is much greater than EFM. The minimum and maximum value of SFDR is 75 dB and 76.53 dB respectively, which show that the deviation of SFDR during the whole conversion process is negligible. Therefore, the next positive point in terms of its stability is that it doesn't degrade its SFDR. While the minimum and maximum value of SFDR of existing EFM architecture is −38 dB and −36 dB, which represents the poor control over the stability issues. Also, the blue curve demonstrates that the SFDR decreases during the whole conversion process which eventually degrades its SFDR. Hence the proposed architecture provides more stability than the conventional architecture. The simulation results emphasize that it reduces spurious tones and limit cycles in its spectrum quite effectively.

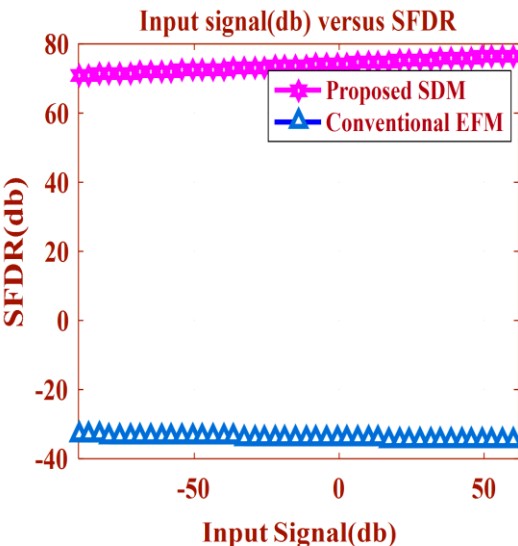

**Figure 16.** SFDR vs. Input signal.

The next parameter which indicates the efficiency of any digital-to-analog converter is signal-to-noise and distortion ratio (SNDR) [38]. The red and blue curves represent the SNDR of EFM and 2-stage feed forward filters based Hybrid MASH-EFM respectively in Figure 17. The maximum SNDR of EFM reaches to 84.56 dB. It can also be seen that the EFM offers degradation in SNDR due to increased noise floor at some higher frequencies. While the proposed architecture achieves the SNDR equals to 85. 24 dB. Once again, dithering proves to be very useful in order to get rid of SNDR degradation by making the noise floor uniform over the whole spectrum.

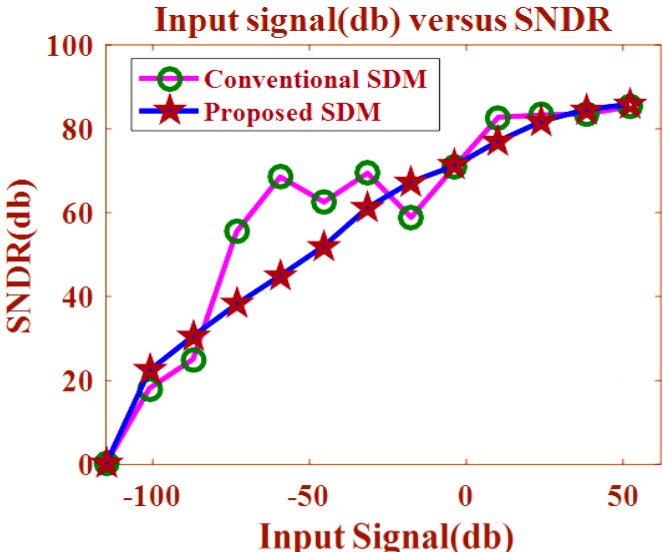

**Figure 17.** SNDR vs. Input signal.

Usually, the overloading effect of the quantizer increases as the ENOB increases. Due to the increase in SNDR in a proposed architecture, the ENOB increases to 14.16. While in the case of 2nd order conventional EFM, the ENOB is 12.20. There is a minor difference between the ENOB of these two architectures. Hence, the proposed architecture achieves outstanding improvement in the performance metrics without overloading the quantizer.

Another main advantage of two-stage feed forward delays and Hybrid MASH-EFM architecture is the use of digital filters and discrete transfer functions. Therefore, this architecture gets rid of complications of mapping of discrete transfer functions to continuous

transfer functions. Hence, this architecture reduces hardware complexity issues due to the use of digital filters.

## 6. Conclusions and Future Recommendations

This work demonstrates that the proposed architecture is effective as compared to existing EFM architecture due to its prominent noise shaping technique. The digital delays are used efficiently to prevent the quantizer from saturation at the higher input levels. Dithering is effectively used to remove limit cycles, idle tones and restricting the noise floor uniform over the entire spectrum. Similarly, the quantization noise follows the NTF curve, which reveals that the proposed architecture does not behave non-linearly at the higher input levels. The SQNR reaches to 86.68 db which shows significant improvement than conventional EFM, which attains SQNR of 75.25 dB, due to the better suppression of quantization noise in the signal bandwidth. Similarly, the superiority of the proposed architecture can also be observed by the improvement of SNDR value up to 85.24 dB, which is higher than the SNDR of the conventional EFM, while the conventional EFM obtains SNDR of 84.56 dB.. The other performance parameters are also super-eminent as illustrated in Table 1. The proposed architecture also avoids the mapping of continuous transfer functions to discrete transfer functions as it is completely designed in the z-domain. The other main aspect of the proposed architecture is that the ENOB achieved by the third-order Hybrid MASH-EFM is below 15 (not high), which ensures less transistor count because of using only one bit quantizer and digital loop filters. This modulator gains high performance metrics with greater stability for both time varying and non-time varying inputs. Because it avoids the overflow conditions which make it more robust.

In future, the new proposed SDM architecture will be implemented on hardware. Digital filters will consist of integrators, which can easily be employed using adders and the feedback path. The delays will be implemented using counters. Similarly, 1-bit quantizer will consist of only few comparators. So, system level designing of this architecture will require some digital logic functions. The next work will involve its FPGA implementation.

**Author Contributions:** Conceptualization, K.I., M.A., M.A.B. and W.T.T.; methodology, K.I., M.A., M.A.B. and W.T.T.; software, K.I., M.A. and M.A.B.; validation, K.I., M.A., W.T.T., M.A.B., U.A., S.R.A. and M.I.; formal analysis, K.I., M.A., W.T.T., M.A.B., U.A. and M.I.; investigation, K.I., M.A., W.T.T., M.A.B., U.A. and M.I.; resources, K.I., W.T.T., U.A., M.I., I.H., Y.Y.G., F.A.A. and M.R.A.; writing—original draft preparation, K.I., M.A., M.A.B. and W.T.T.; writing—review and editing, K.I., W.T.T., U.A., M.I., I.H., Y.Y.G., F.A.A., M.R.A. and S.R.A.; visualization, K.I., M.A., M.A.B., W.T.T., U.A., M.I., I.H., Y.Y.G., F.A.A. and S.R.A.; supervision, M.A. and M.A.B.; project administration, K.I., M.A., M.A.B., W.T.T., U.A., M.I., I.H., Y.Y.G., F.A.A., M.R.A. and S.R.A.; funding acquisition, W.T.T., U.A., M.I., I.H., Y.Y.G., F.A.A. and M.R.A. All authors have read and agreed to the published version of the manuscript.

**Funding:** Researchers Supporting Project Number (RSPD2023R576), King Saud University, Riyadh, Saudi Arabia.

**Acknowledgments:** Researchers Supporting Project number (RSPD2023R576), King Saud University, Riyadh, Saudi Arabia.

**Conflicts of Interest:** The authors declare no conflict of interest.

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
