# Peer review of "A New Noise Shaping Approach for Sigma-Delta Modulators Using Two-Stage Feed-Forward Delays and Hybrid MASH-EFM"

_electronics, doi:10.3390/electronics12030740_

Round 1
Reviewer 1 Report
|
Conclusion |
To be revised with numeric results. |

Author Response
Cover Letter:
Response to the Reviewer 1
We would like to thank the honorable editor and the reviewers for their excellent suggestions and comments. Without their efforts, our manuscript would not have been in its current shape and form. We have carefully revised the manuscript by considering all the valuable suggestions and comments of the respected reviewers. The suggestions given by the reviewers guided us to address and enhance the technical aspects of the manuscript. We hope that the current revised manuscript will meet the standards of the journal and reviewers. The detail rebuttal is given below.
Reviewer # 1
We highly appreciate the esteemed reviewer for his profound observations and valuable comments. We also highly appreciate the time and effort of the reviewer for reviewing the paper in depth and recommending the invaluable suggestions and corrections.
Comment # 1
A New Noise Shaping Approach for Sigma-Delta Modulators using Two-stage Feed-forward Delays and Hybrid MASH-EFM
This work proposes a new noise transfer function which is a combination of transfer functions of two-stage Feed-forward delays and a novel Hybrid multi-stage noise shaping-error feedback sigma-delta modulator. The noise transfer function of two-stage Feed-forward delays mitigates the concern of quantizer saturation. The noise transfer function offered by the Hybrid multi-stage noise shaping-error feedback architecture provides sustainable solutions to limit cycles and idle tones. The simulation concludes that the proposed noise shaping approach obtains comparatively high signal-to-quantization noise ratio than the conventional error feedback modulators. Other performance parameters like spurious-free dynamic range, effective number of bits and signal-to-noise plus distortion ratio are also significantly improved. The other main aspect of this architecture is that it ensures less transistor count because of using only one bit quantizer and digital loop filters. This modulator gains high performance metrics with greater stability for both time varying and non-time varying inputs.
Rebuttal
The authors are thankful to the reviewer for his valuable suggestions. According to the reviewer comments, we have revised the Conclusion with the numeric results. The changes can be tracked in the Conclusion section. Please see Section 6, Conclusions and Future Recommendations. Please see Page 15, Line No. 461 – 468 and Page 15, Line No. 470 – 475 of the manuscript.

Reviewer 2 Report
The manuscript deals with signal filtering with the purpose to improve dynamic range, effective number of bits and signal-to-noise ofn high-performance analog-to-digital and digital-to-analog converters. Paper presents some novel materials and may be of interest for readers who design and use of high performance DAC and ADC in such fields as industrial processes control instrumentation, seismic activity measurements, speech and video signal processing, design of Integrated System digital Network, digital cellular radio, mobile communication devices and other applications. The materials are well organized and provide adequate description of the propose methods and results of the simulation. Meanwhile there are some drawbacks lacks of information which should be overcame before publishing.
1. Terms MASH and ADM appear in Introduction immediately after eq.(1) with the following analysis of the problems associated with t MASH and ADM technology utilization. Meanwhile explanation of MASH and ADM principles is given in the next chapter. It is not convenient for reader and it would be reasonable if authors add few words description of how MASH and ADM work immediately after introducing these terms.
2. Notations used in Figure 1 are not explained. Expalnation should be added in associated text.
3. Terms CT/ DT are not explained.
4. Some of the notations used in chapter 3 are not explained: “?3[?]” and “?[?]” in eq. (1) (besides, what does index “n” mean?), “D(z)” in eq. (6), “A” in eq. (11)
5. Chapters 3 and 4 have the same title “A New Proposed Noise Transfer Function”. Should be modified.
6. Term SFDR is not explained. Should be added.
Author Response
Cover Letter:
Response to the Reviewer 2
We would like to thank the honorable editor and the reviewers for their excellent suggestions and comments. Without their efforts, our manuscript would not have been in its current shape and form. We have carefully revised the manuscript by considering all the valuable suggestions and comments of the respected reviewers. The suggestions given by the reviewers guided us to address and enhance the technical aspects of the manuscript. We hope that the current revised manuscript will meet the standards of the journal and reviewers. The detail rebuttal is given below.
Reviewer # 2
We highly appreciate the esteemed reviewer for his profound observations and valuable comments. We also highly appreciate the time and effort of the reviewer for reviewing the paper in depth and recommending the invaluable suggestions and corrections.
Comment # 1
Terms MASH and ADM appear in Introduction immediately after eq.(1) with the following analysis of the problems associated with t MASH and ADM technology utilization. Meanwhile explanation of MASH and ADM principles is given in the next chapter. It is not convenient for reader, and it would be reasonable if authors add few words description of how MASH and ADM work immediately after introducing these terms.
Rebuttal
The authors are thankful to the reviewer for his valuable suggestions. According to the reviewer comments, we have added the explanation about MASH architecture immediately after the equation 1 in the Introduction section and removed this redundant information in the section 2. Please see Section 1, Introduction and Section 2, Related Work. Please see Page 2, Line No. 87 – 93 and Page 3, Line No. 137 – 141 of the manuscript.
Comment # 2
Notations used in Figure 1 are not explained. Explanation should be added in associated text.
Rebuttal
The authors are again thankful to the reviewer for his valuable suggestion and value addition. The authors have explained the notations used in Figure 1. Please see Section 2, Related Work. Please see Page 4, Line No. 153-156.
Comment # 3
Terms CT/ DT are not explained.
Rebuttal
The authors are again thankful to the reviewer for his valuable suggestion and value addition. As per suggestion, the authors have modified accordingly. Please see Section 2, Related Work. Please see Page 4, Line No. 171-172.
Comment # 4
Some of the notations used in chapter 3 are not explained: “?3[?]” and “?[?]” in eq. (1) (besides, what does index “n” mean?), “D(z)” in eq. (6), “A” in eq. (11)
Rebuttal
The authors are again thankful to the reviewer for his valuable suggestion and time. Please see Section 3, A New Proposed Noise Transfer Function. Please see Page 5, Line No. 203-204. Please visit Page 6, Line No. 223 and Page 6, Line No. 246.
Comment # 5
Chapters 3 and 4 have the same title “A New Proposed Noise Transfer Function”. Should be modified.
Rebuttal
The authors are again thankful to the reviewer for his valuable suggestion and value addition. The authors have modified the manuscript accordingly. Please see Section 4, Simulation Methodology. Please visit Page 8, Line No. 269.
Comment # 6
Term SFDR is not explained. Should be added.
Rebuttal
The authors are again thankful to the reviewer for his valuable suggestion and value addition. The authors have modified the manuscript accordingly. Please see Section 5, Experimental results and Discussion. Please visit Page 13, Line No. 419-420.

Round 2
Reviewer 2 Report
Thank you for manuscript editing.
The revised version can accepted for publication.